# Precision Phenotyping of Agro-Physiological Responses and Water Use of Sorghum under Different Drought Scenarios

Thierry Klanvi Tovignan [1,2,3,*](ID), Yasmeen Basha [2], Steffen Windpassinger [2], Sruthy Maria Augustine [2], Rod Snowdon [2] and Stjepan Vukasovic [2]

1 Département de Génétique et de Biotechnologie, Faculté des Sciences et Techniques (FAST), Université d'Abomey-Calavi (UAC), Cotonou BP 1947, Benin
2 Department of Plant Breeding, Research Center for BioSystems, Land Use and Nutrition (IFZ), Heinrich-Buff-Ring 26-32, 35392 Giessen, Germany
3 Centre Haïtien d'Innovation en Biotechnologies et pour une Agriculture Soutenable (CHIBAS), Faculté des Sciences de l'Agriculture et de l'Environnement, Université Quisqueya, Port-au-Prince BP 796, Haiti
* Correspondence: thierry.tovignan@uniq.edu

**Abstract:** Understanding sorghum response to water stress at different developmental stages is important for developing sorghum varieties with improved tolerance to drought. This study set out to finely characterize key agro-physiological responses and water use of sorghum subjected to different drought scenarios. A greenhouse experiment was conducted using a DroughtSpotter facility that enables real-time quantification of water use by gravimetric tracking. Four different water treatments were assessed: (i) early vegetative drought stress (DS1), (ii) DS1 plus recovery (DS1R), (iii) late vegetative drought stress (DS2), and (iv) well-watered (WW). Plant pheno-morphology and yield data were recorded. Leaf fluorescence and photosynthesis were continuously recorded throughout the experiment. Our results showed that green leaf number and plant leaf area were more affected by DS2 than DS1 and delayed plant flowering. Nevertheless, plants in DS2 were taller and produced higher stem and leaf biomass compared to DS1. No significant difference was recorded in grain yield between DS1 and DS2 but were outperformed by DS1R and WW. The transpiration and photosynthetic rates were shown to decrease at the same time as the stomatal conductance. This can therefore be assimilated to a stomatal down-regulation limiting $CO_2$ uptake. However, the increase in intercellular $CO_2$ concentration is likely to indicate the presence of $CO_2$ in the substomatal cavity that was not conveyed to the carboxylation sites. This suggests a non-stomatal limitation of the photosynthesis. Moreover, the plants recovered quite well from DS1, and this was more prominent for physiological parameters than morphological ones. Globally, water use efficiency (WUE) for DS2 was higher compared to WW and DS1 treatments, confirming the growing point differentiation as a critical stage where drought stress should be avoided to ensure yield and better WUE. Adaptation responses were related to the reduction of transpiration through plant leaf area reduction, the reduction of stomatal conductance, and the increase of intercellular $CO_2$ limiting photosynthesis. Further studies focusing on the biomarkers of stress and transcriptomic analyses are needed to provide further insight into the drought adaptation mechanisms of this line.

**Keywords:** sorghum; drought adaptation mechanisms; photosynthesis; transpiration; water use efficiency; drought recovery

## 1. Introduction

Sorghum is an important cereal crop in both dryland areas of Africa and India, where it is used as a staple crop for animal feed and as a source of income for small-scale farmers [1], and in Australia, the United States of America, and South America, where it is grown as a commodity crop for animal feed, bioethanol, or export. In contrast, in temperate Europe, it is a novel crop with minor importance that is mainly used for biogas production [2].

Sorghum as an annual crop can be grown in low rainfall environments, which makes it a good alternative to maize and sugarcane in these areas. Its genetic diversity and spectrum of adaptation to various cropping environments suggests considerable opportunity for improvement from the perspectives of diversifying end uses (in terms of biomass or grain) and adaptation to climate change [3]. As a C4 species, it has a high potential for producing biomass for different end uses depending on the range of biochemical composition available within its genetic diversity: digestible, poorly lignified biomass for biofuel, biogas, and forage, or lignified low sugar for bioplastic and bio-concrete [4,5]. Part of its value resides in this ability to be a multipurpose crop, which is a key advantage for optimizing land and resource uses and contributing to agriculture sustainability [6].

However, this sort of multipurpose production, which requires optimal access to light, water, and mineral nutrients, could be compromised when access to these resources is limited, mainly in the context of climate variability, which accentuates the scarcity of water resources. In particular, drought stress can negatively affect sorghum development and yield (e.g., biomass, grain) depending on its intensity and the stage at which it occurs. Drought persistence was shown to weaken sorghum plants and favor diseases or fungal infestation, leading to grain yield loss by 20 to 60% [7]. Occurrence of drought before anthesis can also reduce stem biomass production by 42% [8]. In some instances, drought causes a total failure of the crop [9].

Drought tolerance in sorghum involves the interaction between different morphological structures, biochemical expressions, and physiological functions [10]. Early drought stress is reported to induce changes in sorghum physiology by decreasing stomatal conductance, which can lead to the reduction of $CO_2$ uptake and the leaf transpiration rate [11]. It has also been shown to reduce leaf water potential and the maximum quantum efficiency of photosystem II [12,13]. However, the leaf area temperature was found to be increased under drought [14]. As a result, early drought stress affects plant photosynthesis and in fine biomass and grain production. Plant photosynthetic activity can also be disrupted when drought stress causes damage to cell membranes [15]. Therefore, genotypic ability to stabilize cell membranes is a promising way to enhance drought tolerance in crops [16]. Moreover, rewatering after drought stress contributes to recovering the performance of the plant. Gano et al. [11] researched a panel of ten West African sorghum varieties and reported good recovery for the number of appeared leaves and photosynthesis rate, whereas plant height and biomass production were hardly recovered. Jedmowski et al. [13] observed good recovery for leaf relative water content, whereas the recovery of leaf fluorescence parameters (Fv/Fm and performance index) were genotype-dependent. Moreover, Martínez-Goñi et al. [12] showed that plants prioritized the recovery of the net photosynthetic rate under ambient $CO_2$ compared to elevated $CO_2$. Therefore, since there is still a knowledge gap on sorghum recovery from drought stress after rehydration, there is a need for more investigation.

Understanding drought adaptation mechanisms is important for developing drought-tolerant sorghum varieties. The present work aimed at finely characterizing key agro-physiological responses and water use of sorghum subjected to different pre-flowering drought scenarios. The effect on plant structural growth and development, biomass, and grain production were assessed, as well as the anatomo-physiological mechanisms underlying drought adaptation in sorghum. In contrast to post-flowering drought stress in sorghum, which has been shown to be associated with stay green (e.g., Borrell et al. [17]), pre-flowering drought stress has received less attention. This may be due to the fact that post-flowering or terminal drought stress is of higher relevance globally, since at this stage, soil water tends to be depleted, while earlier vegetative growth in high-quality soils still benefits from stored water and is less likely to be drought-affected. However, at shallow soils with low water capacity (which are, e.g., typical sorghum cropping environments in temperate Europe), drought stress can occur at early stages. Pre-flowering and post-flowering drought tolerance in sorghum are believed to be based on different mechanisms [18], and usually, a sorghum genotype shows tolerance against only one of



them [19]. Hence, the goal of this study was to provide more insight into the physiological responses to both of these drought stress regimes.

## 2. Materials and Methods

The current study was a greenhouse experiment conducted from September 2021 to April 2022 at the Department of Plant Breeding of Justus Liebig University of Giessen (Germany).

### 2.1. Plant Material

The sorghum genotype used was SC101, a conversion line [20] of caudatum–kafir race. It was chosen for this study from the sorghum collection of the Plant Breeding Department of Justus Liebig University of Giessen due to its observed superior drought tolerance in field experiments.

### 2.2. Methods

2.2.1. Experimental Conditions

The experiment was conducted as a full-growth cycle trial using the DroughtSpotter® system (Phenospex, Heerlen, The Netherlands) facility located at University of Giessen, Hesse, Germany. The DroughtSpotter® is a phenotyping platform designed for drought-stress-related experiments using growth containers placed on weight scales, which record weight deviations every five minutes throughout the whole experiment [21]. Further, every container is individually connected to an irrigation system, allowing specific irrigation treatment for each growth container of 60 L filled with 80 kg soil medium. The potting media were composed of 40% of excavated soil from a local field and 60% sand to ensure sufficient drainage throughout all soil layers. The soil texture and nutrient contents are presented in Table 1. The soil pH was 7. The soil texture was mostly sandy: fine sand (31.5%), middle sand (25.0%), large silt (20.6%), and clay (8.3%).

**Table 1.** Texture and nutrients contents of the soil used to fill the pots.

| Analysis of Soil Texture | | | |
|---|---|---|---|
| Type | Size [mm] | Unit | Value |
| Fine sand | 0.063–0.2 | % | 31.50 |
| Middle sand | 0.2–0.63 | % | 25.00 |
| Large silt | 0.02–0.063 | % | 20.60 |
| Clay | <0.002 | % | 8.30 |
| Middle silt | 0.0063–0.02 | % | 7.50 |
| Fine silt | 0.002–0.0063 | % | 3.70 |
| Large sand | 0.63–2 | % | 3.50 |
| **Soil nutrient contents** | | | |
| Element | Symbol | Unit | Value |
| Phosphorus | $P_2O_5$ | [mg/100 g] | 14.00 |
| Potassium | $K_2O$ | [mg/100 g] | 8.00 |
| Magnesium | Mg | [mg/100 g] | 10.00 |
| Iron | Fe | mg/kg | 76.40 |
| Copper | Cu | mg/kg | 1.44 |
| Zinc | Zn | mg/kg | 1.75 |
| Manganese | Mn | mg/kg | 37.40 |
| Boron | B | mg/kg | 0.17 |
| Molybdenum | Mo | mg/kg | <0.0150 |

2.2.2. Experiment Design and Management

A complete randomized design was used to test four water treatments, which included (i) early vegetative drought stress (DS1), (ii) DS1 plus recovery (DS1R), (iii) late vegetative drought stress (DS2), and (iv) well-watered treatment (WW). The experiment was conducted using two replications (i.e., containers per treatment). In each container,



sorghum seeds were sown in three positions, which were located equidistant from each other. Three to four seeds were sown per position. At seedling stage, the plants were thinned to one per position, and three plants were maintained per container. Irrigation was provided every day at midnight following an automatic schedule.

Figure 1 depicts the cumulative transpiration showing how irrigation was managed for the different treatments. At physiological maturity, the cumulated water supplied to DS1, DS1R, and DS2 was 50%, 67%, and 61%, respectively, of the cumulated water supplied to the well-watered treatment (the control). In fact, DS1 was the first stress applied at 48 days after sowing, when the plants had 8 leaves and were at the growing-point differentiation stage (ca. BBCH 30; BBCH stands for *Biologische Bundesanstalt*, *Bundessortenamt und Chemische Industrie*, a scale used to identify the phenological stage of the plant). It was a decreased irrigation corresponding to 25% of field capacity. It lasted 63 days and thereafter, it was maintained at 35% of field capacity until physiological maturity. DS1R consisted of DS1 treatment, which after 63 days under stress (the plants had 15 leaves), was followed by optimal rewatering (70% of field capacity) until physiological maturity. DS2 was the treatment that was well-watered (70% of field capacity) until 118 days after sowing, and afterwards, irrigation was decreased (when the plants had 20 leaves) so as to maintain the treatment at 35% of field capacity until physiological maturity. As for the well-watered treatment, it was maintained under optimal irrigation (70% of field capacity) throughout the experiment.

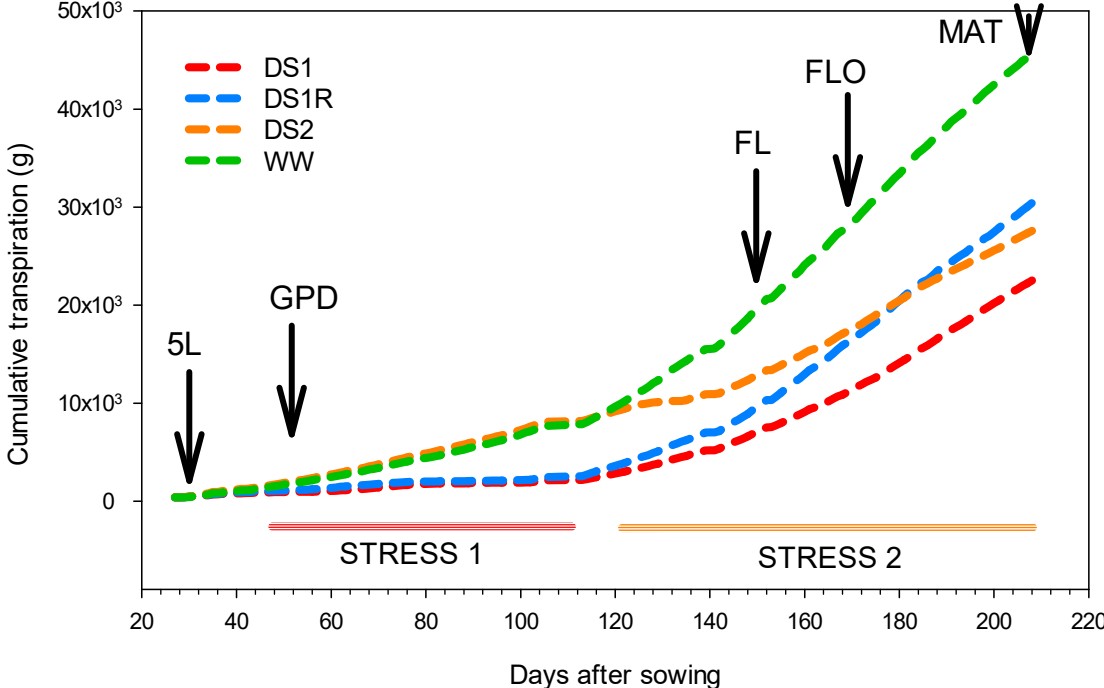

**Figure 1.** Cumulative transpiration compared between different water regimes. 5L: five leaf, GPD: growing point differentiation, FL: flag leaf, FLO: flowering, MAT: physiological maturity.

As for the fertilizer application, a WUXAL Super (AGLUKON Spezialduenger GmbH & Co. KG, Düsseldorf, Germany) nutrient solution was applied as recommended by the manual to exclude any nutrient deficiencies distorting the experiment. A volume of 50 mL was provided per position and 150 mL per container once a week. Fertilizer application started from the 20th day after sowing and was not applied when irrigation was withheld to impose stress so as to avoid additional water supply.

### 2.2.3. Weather Conditions

Temperature and relative humidity were recorded every ten minutes. Both parameters were used to compute the vapor pressure deficit (VPD) using the following formulae used by Alduchov et al. [22]:

$$VPD = SVP \times \left(1 - \frac{RH}{100}\right)$$

where *SVP* is the saturated vapor pressure, computed as:

$$SVP = 610.78 \times e^{\frac{-T}{(T+273.3)} \times 17.2694}$$

*T* is the temperature (°C), and *RH* is the relative humidity (%).

During the experiment, the temperature was 24.1–26.5 °C during the day and 18.1–23.8 °C during the night (Figure 2). These are moderate temperatures for sorghum, meaning that drought stress was not accompanied by high temperature stress, as it frequently occurs in its cropping environments. Inversely, the relative humidity was at 47.7–51.8% during the day and 48.4–55.3% during the night. As for VPD, it was 1.54 to 1.71 during the day and 0.93–1.53 during the night. The temperature remained almost similar over the duration of the experiment, while the relative humidity was fluctuating along the experiment.

The day/night regime was 16/8 h.

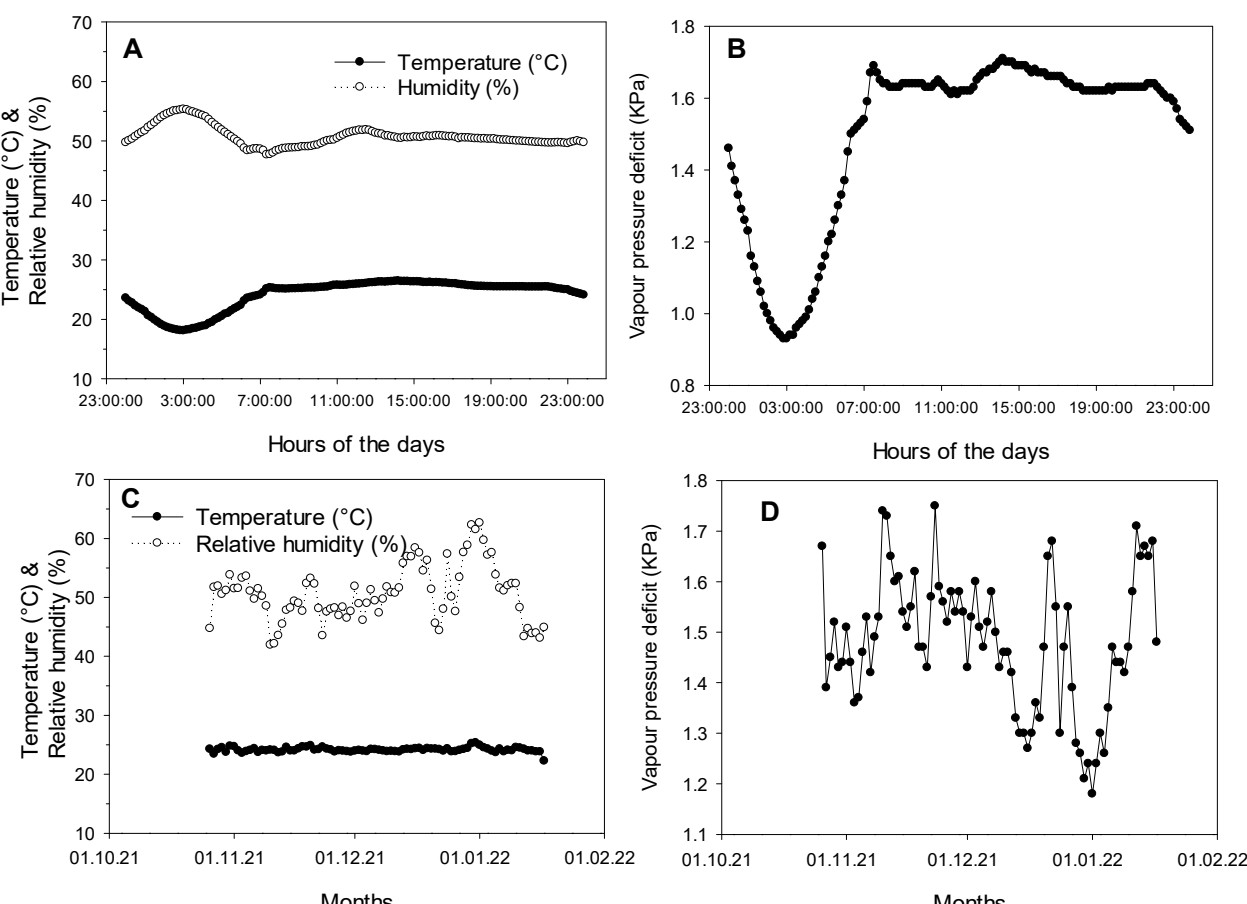

**Figure 2.** Daily averages of temperature and relative humidity (**A**), daily average vapor pressure deficit (**B**), and the temperature and relative humidity (**C**), and vapor pressure deficit (**D**) recorded along the experiment.

*2.3. Data Collection*

2.3.1. Plant Phenology and Morphology

At thinning, the three plants retained per container were tagged for dynamic characterization of plant growth and development. The appeared and ligulated leaves were counted three times per week. The green leaves were also counted. The leaf length and the largest part of leaf width were measured and were used to estimate the leaf area as the product of length, width, and allometric coefficient (0.69) [23]. Since the measurements were done thrice a week, the dimensions of all the leaves were taken up to the flag leaf. The total plant leaf area was computed as the sum of the area of the green leaves at the given stage. The phyllochron was computed as an inverse of development rate, calculated using the appeared leaf number as described in Tovignan et al. [24]. The phyllochron is the time that elapses between the appearance of two successive leaves.

Plant height was also measured dynamically up to flag leaf stage, as the height from the soil surface to the ligule of a given ligulated leaf. At physiological maturity (MAT), the plant height was also measured now from the soil surface to the tip of the panicle. The stem median diameter was measured at MAT using a digital Vernier caliper. The stem elongated internodes were counted. Peduncle length and panicle length and width were also taken at MAT.

The flag leaf and flowering are the phenological stages that were recorded. These stages were noted when 50% of the plants of each treatment reached the given stage.

2.3.2. Biomass Production

At physiological maturity, the three plants of each container were harvested and fresh stem and leaf weights were measured. The dry weights were estimated once stem and leaf biomass were dried in an oven at 70 °C during one week.

2.3.3. Grain Production

The panicle fresh and dry weight were recorded in grams in both control and stressed plants using a digital balance. The panicle fresh weight was measured immediately after harvesting of the plants. For the dry weight, the panicles were kept in oven for one week at 70 °C. Then, grain weight per panicle (GWP), one hundred seed weight (P100), and grain number per panicle (GNP) were taken.

2.3.4. Water Use Efficiency

Water use efficiency was estimated at physiological maturity as the ratio between the biomass produced and the cumulative transpiration. It was determined for stem fresh and dry weights, leaf fresh and dry weights, and grain production traits such as panicle fresh and dry weights and grain weight per panicle.

2.3.5. Gas Exchange, Chlorophyll Fluorescence, and Chlorophyll Content

Plant photosynthesis and leaf fluorescence were recorded three times per week from the setup of the first drought stress using LI-6800 (LICOR Biosciences, Lincoln, NE, USA). These measurements were taken on the third leaf from the top when the plants were photosynthetically active (between 9:00 a.m. and 1:00 p.m.), i.e., when the stomata are supposed to be well-opened, with the photon flux density fixed at 1000 μmol m$^{-2}$ s$^{-1}$, and a mean temperature of 23 °C; the mean $CO_2$ concentration was 400 μmol mol$^{-1}$ air, and the pump flow was 600 μmol s$^{-1}$.

Plant photosynthesis parameters such as net photosynthetic rate, transpiration rate, stomatal conductance (to water and $CO_2$), and intercellular $CO_2$ concentration were recorded. Instantaneous water use efficiency was calculated as the ratio between net photosynthetic rate and transpiration rate.

For the leaf fluorescence, a maximum yield of PSII (Fv/Fm) was used in this study.

Chlorophyll content was also measured three times per week using a hand-held chlorophyll content meter (CCM 200 plus, Opti-Sciences, Hudson, NH, USA).

2.3.6. Drought Recovery Index

Drought recovery index was calculated for the studied traits using the following formula:

$$DRI = \log A + 2 \log B.$$

where A is the relative trait measured at the end of the drought and B is the relative trait measured 2 weeks after re-watering.

2.3.7. Leaf Anatomical Structures Analysis

In order to assess the percentage of cell membrane injury caused by drought stress, leaf samples were collected from the third leaf from the top, from well-watered and drought-induced treatments at the end of early vegetative drought stress. Sorghum leaf discs (~10 × 5 mm) were stained with AlexaFluor 488 phalloidin as previously described [25] with slight modifications [26]. The discs were fixed in 3.5% (*v*/*v*) formaldehyde in phosphate-buffered saline (PBS, pH 7.4) at room temperature overnight. After washing in PBS, the discs were immersed in 0.5% (*v*/*v*) Triton X-100 in PBS (pH 7.4) at room temperature overnight. The discs were then washed three times in PBS and stained with 0.66 mM AlexaFluor 488 phalloidin in PBS (Thermo Fisher Scientific, Waltham, MA, USA) at room temperature for 1 h in the dark before rinsing in PBS and mounting in PBS on glass slides. The interlocking marginal lobes were observed under a confocal microscope.

*2.4. Data Analysis*

One-way ANOVA analysis was conducted to test the effect of water regime on plant pheno-morphological and physiological data collected. The ANOVA was followed by a Tukey HSD test for mean comparison. Standard error was calculated to assess the degree of variation around means. ANOVA analysis and mean comparison test were performed using R 4.1.2 [27].

The leaf development rate of the plants in each water treatment was determined by assessing the slopes of the multi-regression lines between the appeared leaf number and the days after sowing. For this purpose, piecewise regression was used to detect the breakpoint at which the rate changed. Piecewise regression analysis was performed using SigmaPlot 14.0 (Systat Software, Inc., Los Angeles, CA, USA).

**3. Results**

*3.1. Effect of Different Drought Scenarios on Plant Phenology*

Drought stress applied has significantly affected plant phenology (Figure 3). The plants in drought conditions reached flag leaf (FL) and flowering (FLO) later than those grown under well-watered conditions. The flag leaf ligulation was delayed for DS1 and DS2 by 9 and 21 days, respectively, while the flowering time was delayed by 2 and 7 days for DS1 and DS2, respectively.

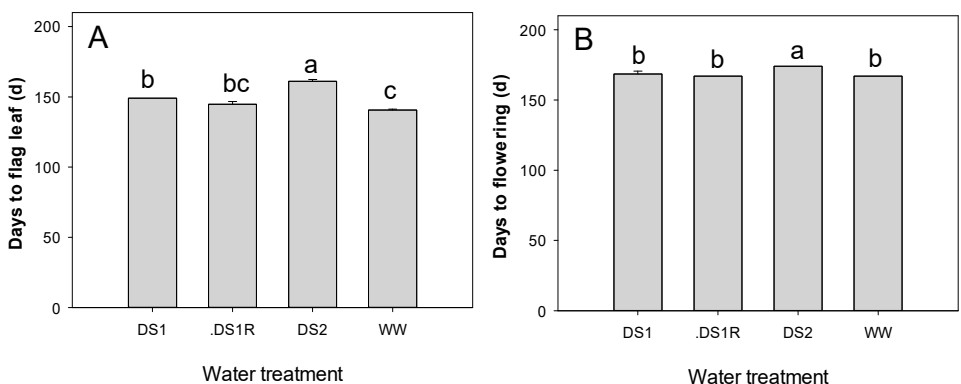

**Figure 3.** Days to flag leaf (**A**) and flowering (**B**) compared among the water treatments. Among water treatments, the traits mean with different letters are statistically different (*p* < 0.05).

### 3.2. Effect of Different Drought Scenarios on Plant Growth and Development

The effect of drought stress on plant morphology varied depending on the phenological stage. Plant height (PH) evolution compared among different water treatments (Figure 4E) revealed a 25% reduction for DS1 compared to well-watered treatment at 76 days after sowing. Once irrigation was resumed, the plants recovered from the first stress and showed a gain in height of 20%. After the induction of the second stress (DS2), a drastic reduction of 35% compared to well-watered (the control) was noticed on plant height at 149 days after sowing.

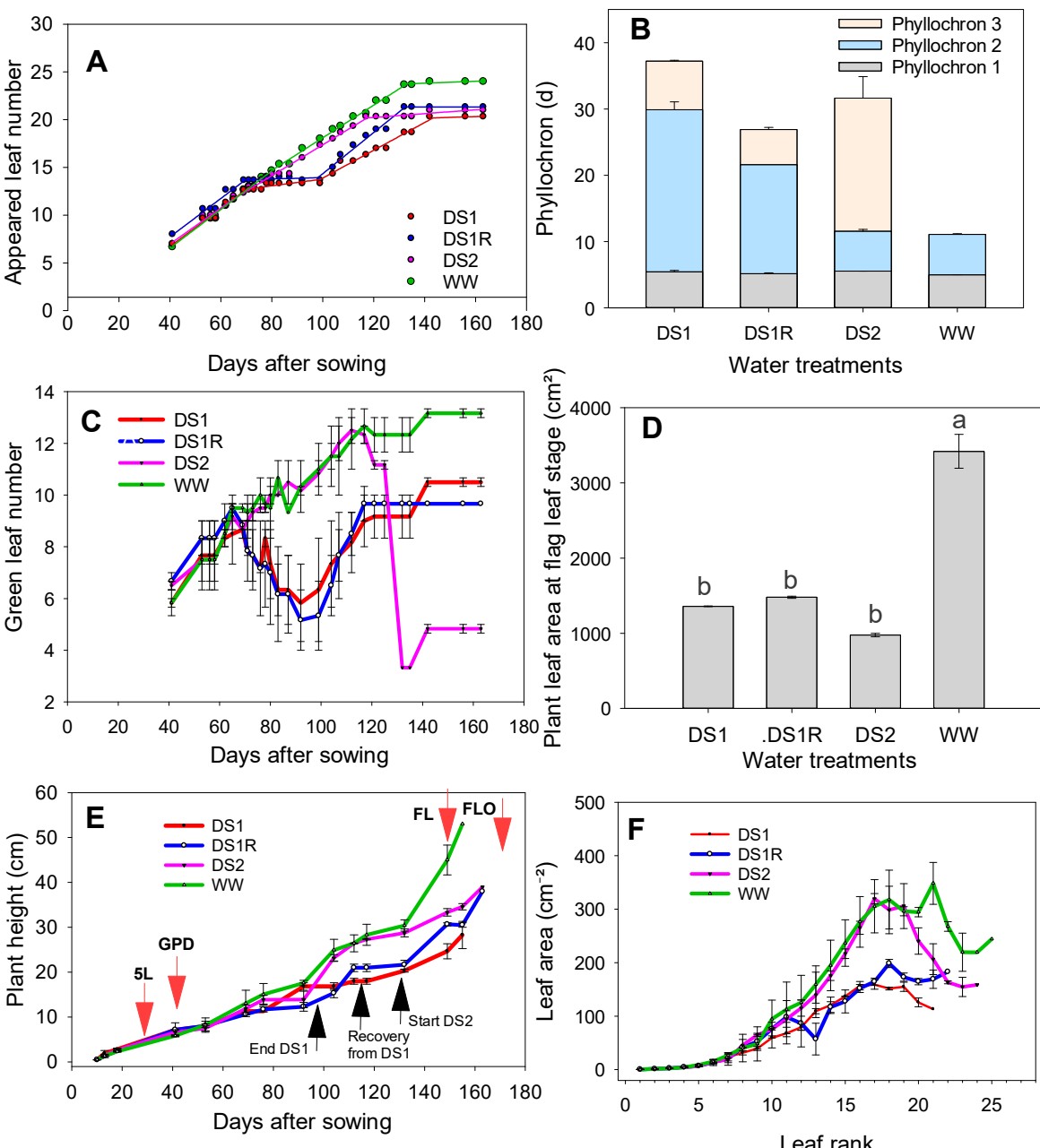

**Figure 4.** Leaf appearance rate (**A**), phyllochron (**B**), green leaves evolution (**C**), plant leaf area at flag leaf stage (**D**), plant height growth rate (**E**), and leaf area per leaf rank (**F**) compared among different water treatments. 5L: five leaf, GPD: growing point differentiation, FL: flag leaf stage, FLO: flowering. For plant leaf area at flag leaf stage, among water treatments, the mean values with different letters are statistically different ($p < 0.05$).

The development rate computed using the appeared leaf number showed three phases for the droughted treatments versus two for the well-watered treatment (Figure 4A,B). At the start, in the absence of any stress, a much faster first phase was observed for all the treatments, where the plant took an average of 5 days to produce a leaf. When the first stress occurred, the leaf production rate was completely slowed down, and the plant took an average of 20 days to produce a leaf (DS1 and DS1R). When irrigation was resumed for DS1R, there was a resumption of growth with a slightly more accelerated rate of 5 days to produce a leaf. On the other hand, for DS2 treatment, which had a normal growth before the stress induction, the leaf emergence was completely slowed, and the plant took almost 20 days to produce and develop the remaining leaves. In contrast, the irrigated treatment showed only two phases: a much faster first phase, wherein the plant took 5 days to produce a leaf, and a slightly slow second phase, wherein it took 6 days to produce a leaf.

The leaf area of the different leaf rank estimated showed more increased leaf expansion under well-water conditions compared to droughted ones (Figure 4F). At the flag leaf stage, DS2 experienced severe drought stress, and this significantly affected its plant leaf area compared to well-watered treatment (WW) (Figure 4D). Late vegetative drought stress led to higher reduction of plant leaf area at the flag leaf stage than early vegetative drought stress. This reduction of the plant leaf area resulted from the reduction of green leaves (Figure 4C). The DS2 treatment, which had time to set up the majority of its leaves and therefore had a great need for water, had to get rid of the majority of its leaves to reduce transpiration. This suggests an adaptation strategy to cope with late vegetative drought stress.

The way early drought stress affected the plant morphology is shown on Figure 5, particularly by reducing plant height, appeared leaf number, and the size of the leaves.

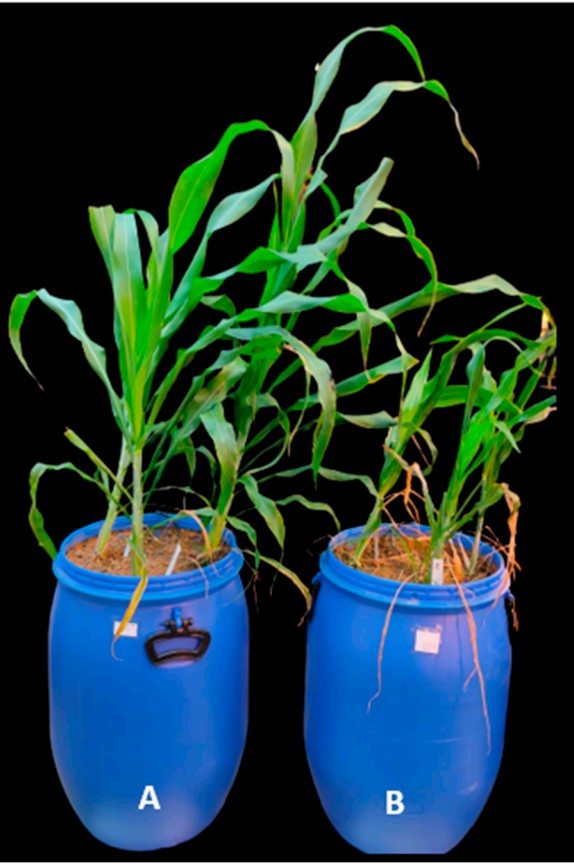

**Figure 5.** Picture showing plants in well-watered (**A**) vs. early drought stress (**B**) conditions 38 days after drought stress induction.

### 3.3. Effect of Different Drought Scenarios on Sorghum Agro-Morphological Parameters Measured at Physiological Maturity

Among the plant morphological traits assessed, water regime has significantly affected stem diameter and plant height when comparing drought stress treatments to well-watered (Table 2). However, the effect was slightly more depressive with DS1 than DS2. The length of peduncle was also affected but the effect was more depressive with DS2.

**Table 2.** Plant morphology, stem and leaf biomass, and grain production at physiological maturity, compared among different water treatments.

|  | DS1 | DS1 R | DS2 | WW | *p*-Value |
|---|---|---|---|---|---|
| **Plant morphology** | | | | | |
| Diam (mm) | 6 ± 0.3 b | 7.5 ± 1.3 b | 9.1 ± 0.9 ab | 11.7 ± 1.4 a | 0.0164 * |
| PH (cm) | 82.6 ± 7.6 b | 94.3 ± 1.8 ab | 82.7 ± 0.5 b | 106.7 ± 0 a | 0.027 * |
| IN | 7.7 ± 0.5 a | 6.7 ± 0.5 a | 6.7 ± 1.4 a | 8.2 ± 1.2 a | 0.36 |
| Lped (cm) | 26.0 ± 1.4 a | 29.3 ± 1.4 a | 19.5 ± 0.7 b | 23.3 ± 0.5 ab | 0.0151 * |
| Lpan (cm) | 17 ± 1.9 a | 19.5 ± 0.2 a | 18.2 ± 0.7 a | 21.5 ± 0.7 a | 0.0805 |
| Wpan (g) | 2.6 ± 3.7 a | 3.3 ± 5.1 a | 2.8 ± 2.2 a | 4.1 ± 5.8 a | 0.0805 |
| **Biomass** | | | | | |
| SFW (g) | 28.1 ± 6.1 b | 43.7 ± 3.1 b | 39.2 ± 1 b | 71.6 ± 4.6 a | 0.00928 ** |
| LFW (g) | 24.1 ± 0.8 b | 31 ± 2.3 b | 30.6 ± 1.3 b | 49.9 ± 2.5 a | 0.00411 ** |
| SDW (g) | 8.5 ± 2.1 b | 12 ± 0.7 b | 13.7 ± 0.4 b | 23.2 ± 1.6 a | 0.00874 ** |
| LDW (g) | 7.6 ± 0.4 b | 9.3 ± 0.8 b | 14.5 ± 2.2 ab | 18.1 ± 1.6 a | 0.0155 * |
| **Grain production** | | | | | |
| PFW (g) | 13 ± 3.3 b | 25 ± 0.8 ab | 15.3 ± 1.8 b | 34.2 ± 4.8 a | 0.0265 * |
| PDW (g) | 9.5 ± 3.1 b | 18.1 ± 0.8 ab | 10.9 ± 1.3 ab | 25.1 ± 4.3 a | 0.0155 * |
| GWP (g) | 8.3 ± 1.9 c | 15.5 ± 0.6 ab | 10.2 ± 1.3 bc | 19.8 ± 0.8 a | 0.0119 * |
| P100 (g) | 1.4 ± 0.1 a | 1.9 ± 0.2 a | 1.2 ± 0.1 a | 1.6 ± 0.2 a | 0.0178 * |
| GNP | 575.6 ± 123.2 b | 833.8 ± 44.1 ab | 867 ± 178.2 ab | 1248.5 ± 54.4 a | 0.0521 |

* $p < 0.05$; ** $p < 0.01$. Diam: stem median diameter, PH: plant height, IN: internode number, Lped: length of peduncle, LPan: length of panicle, Wpan: width of panicle, SFW: stem fresh weight, LFW: leaf fresh weight, SDW: stem dry weight, LDW: leaf dry weight, PFW: panicle fresh weight, PDW: panicle dry weight, GWP: grain weight per panicle, P100: hundred-seed weight, GNP: grain number per panicle. Among water treatments, the traits mean with different letters are statistically different ($p < 0.05$).

Stem (FW and DW) and leaf (FW and DW) biomass were significantly affected by drought stress, but the effect was more detrimental with DS1. This led to the reduction of stem biomass by 62% with DS1 vs. 43% with DS2, while leaf biomass was reduced by 55% and 29% by DS1 and DS2, respectively.

As for grain production, the water regime significantly affected all the grain production traits (PFW, PDW, GWP, P100) except the number of grains per panicle (GNP). DS1 had the most detrimental effect on these traits compared to DS2. Panicle weight was reduced by 62% by DS1 vs. 56% by DS2. Grain weight per panicle was reduced by 58% by DS1 vs. 48% by DS2. P100 was reduced by 13% by DS1 vs. 25% by DS2, while the number of the grains per panicle was reduced by 54% by DS1 vs. 31% by DS2.

### 3.4. Effect of Early and Late Vegetative Drought Stresses on Sorghum Physiology
Gas Exchange, Chlorophyll Content, and Fluorescence

The plant photosynthetic parameters (gas exchange, chlorophyll content, and fluorescence) recorded during the experiment are presented in Figure 6. Overall, a net contrast can be observed between the well-watered and the droughted treatments. Similar trend was observed for the gas exchange parameters such as transpiration rate (Figure 6A), net photosynthetic rate (Figure 6B), and stomatal conductance (water and $CO_2$) (Figure 6C,D), and instantaneous water use efficiency (Figure 6F). During early drought stress, these traits dropped sharply from 72 to 100 days after sowing (das) before increasing again two weeks later with the rewatering. The decrease during early vegetative drought stress was up to −30% for transpiration rate, −43% for net photosynthetic rate, −31% for stomatal conductance to both water and $CO_2$, and −49% for instantaneous water use efficiency. Meanwhile, an increase of 43% was observed for intercellular $CO_2$ concentration (Figure 6E). A similar trend was observed under the late vegetative drought stress between 128 to 139 days, but to a lesser extent. The late vegetative drought stress led to a reduction of transpiration rate by up to −18%, photosynthesis rate by −28%, stomatal conductance (water or

$CO_2$) by $-17\%$, and instantaneous WUE by $-22\%$, while intercellular $CO_2$ concentration increased by 6.4%.

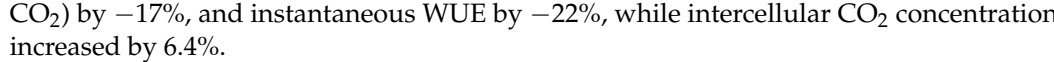

**Figure 6.** Transpiration rate (**A**), net photosynthetic rate (**B**), stomatal conductance to water (**C**), stomatal conductance to $CO_2$ (**D**), intercellular $CO_2$ (**E**), instantaneous water use efficiency (**F**), maximum yield of PSII (**G**), and chlorophyll content (**H**) recorded during the experiment, compared among different water treatments. Presented data are average value with standard error. FL: flag leaf stage, FLO: flowering.

As for leaf fluorescence, the maximum yield of PSII (Fv/Fm) was reduced by $-14\%$ from 80 to 92 days with the first drought stress applied and then increased again by 19% two weeks later with the rewatering. During the late drought stress, it was reduced by 14% from 125 to 132 days.

Chlorophyll content increased for all the treatments between 40 and 72 days. Along with the increasing of the first drought stress, it decreased by $-26\%$ between 69 and 73 days.

After rewatering, it increased again by 76% between 92 and 115 days. During the second drought stress, it decreased again by −28% between 125 and 132 days.

### 3.5. Drought Recovery Index and Water Use Efficiency

Table 3 presents the drought recovery index (DRI) estimated for plant morphological and physiological traits. The smaller the index value for a trait, the lower its ability to recover from drought stress. Traits with better recovery ability had values close to zero or slightly above. Plant height, number of appeared leaves, and intercellular $CO_2$ concentration showed a relatively low recovery ability with DRI of −0.36, −0.17, and −0.20, respectively. The other physiological traits of chlorophyll content, maximum yield of PSII (Fv/Fm), transpiration rate, stomatal conductance to $CO_2$, stomatal conductance to $H_2O$, instantaneous water use efficiency, and photosynthetic rate showed better recovery from drought stress, with DRI ranging from −0.04 to 0.52 as shown in Table 3.

**Table 3.** Drought recovery index of some morphological and photosynthetic traits.

|  | Drought Recovery Index (DRI) |
| --- | --- |
| Plant height, PH | −0.36 |
| Number of appeared leaf | −0.17 |
| Intercellular $CO_2$ | −0.20 |
| Vapor pressure deficit, VPD | −0.04 |
| Chlorophyll content (SPAD) | −0.02 |
| Leaf temperature | 0.02 |
| Maximum yield of PSII (Fv/Fm) | 0.02 |
| Transpiration rate | 0.10 |
| Stomatal conductance to $CO_2$, Gtc | 0.32 |
| Stomatal conductance to $H_2O$, Gsw | 0.32 |
| Instantaneous water use efficiency, iWUE | 0.38 |
| Photosynthetic rate | 0.52 |

The water use efficiency (WUE) is one of the most functional indices that can be used to assess plant optimal water management and its ability to adapt to drought stress. WUE was calculated at physiological maturity as the ratio between the accumulated biomass and the recorded transpiration.

Table 4 presents the comparison of WUE among the different water treatments for stem and leaf biomass and grain production traits. The WUE depended on water regime and also on plant organs. The difference observed among water treatments was only significant ($p < 0.05$) for WUE of stem DW and leaf DW.

**Table 4.** Water use efficiency estimated for stem and leaf biomass and grain production at physiological maturity.

|  | DS1 | DS1R | DS2 | WW | *p*-Value |
| --- | --- | --- | --- | --- | --- |
| Cumulative transpiration (Kg) | 22.8 ± 3.2 a | 30.8 ± 1.5 ab | 27.8 ± 1.9 ab | 46 ± 2.3 b | 0.0486 * |
| Water use efficiency (g Kg$^{-1}$) |  |  |  |  |  |
| SFW | 1.23 ± 0.19 a | 1.42 ± 0.07 a | 1.41 ± 0.03 a | 1.56 ± 0.07 a | 0.413 |
| SDW | 0.42 ± 0.02 ab | 0.39 ± 0.02 b | 0.49 ± 0.01 ab | 0.52 ± 0.01 a | 0.0277 * |
| LFW | 1.06 ± 0.03 a | 1.01 ± 0.05 a | 1.1 ± 0.03 a | 1.09 ± 0.04 a | 0.552 |
| LDW | 0.33 ± 0.01 b | 0.30 ± 0.02 b | 0.55 ± 0.03 a | 0.39 ± 0.03 ab | 0.0163 * |
| PFW | 0.2157 ± 0.1 a | 0.81 ± 0.02 a | 0.55 ± 0.05 a | 0.74 ± 0.07 a | 0.111 |
| PDW | 0.46 ± 0.05 a | 0.59 ± 0.02 a | 0.41 ± 0.02 a | 0.55 ± 0.07 a | 0.227 |
| GWP | 0.39 ± 0.03 a | 0.5 ± 0.01 a | 0.37 ± 0.03 a | 0.43 ± 0.01 a | 0.123 |

* $p < 0.05$. WUE: water use efficiency (g Kg$^{-1}$), SFW: stem fresh weight, LFW: leaf fresh weight, SDW: stem dry weight, LDW: leaf dry weight, PFW: panicle fresh weight, PDW: panicle dry weight, GWP: grain weight per panicle. Among water treatments, the traits mean with different letters are statistically different ($p < 0.05$).

As for grain production traits (PFW, PDW, GWP), no significant differences among the treatments were observed.

### 3.6. Early Drought Effect on Sorghum Leaf Anatomical Structures

The effect of early vegetative drought stress on the leaf anatomical structures is shown in Figure 7. The interlocking marginal lobe (IML) analysis shows that drought-stressed treatment has more IML compared to the well-watered one. In the zoomed area, IML under water stress is more packed in form and greater in number. The frequency of IML under drought stress was more increased compared to well-watered (Figure 7C).

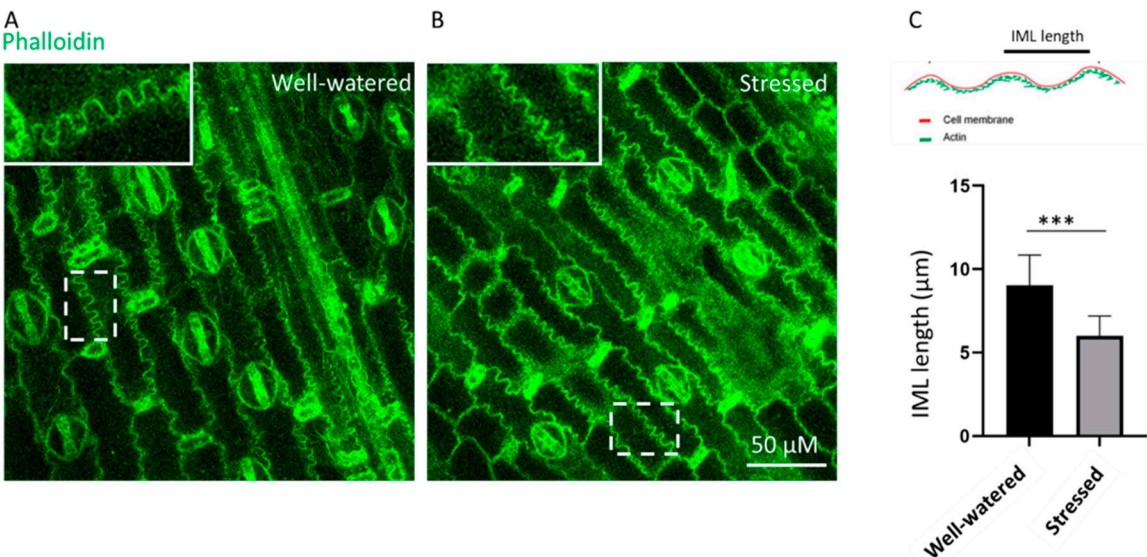

**Figure 7.** Microscopic images of phalloidin-stained sorghum from well-watered conditions (**A**) vs. early vegetative drought-stressed conditions (**B**). Quantification of the length of the IML in well-watered and drought-stressed sorghum (**C**). Data are means ± SD (n = 30 (10 cells)), *** $p < 0.001$.

### 4. Discussion

Drought stress, which is a growing concern in agriculture, can affect crop productivity depending on the development stage at which it occurs. Breeding for more drought-tolerant sorghum lines requires a better understanding of drought adaptation mechanisms (anatomo-physiological) deployed by sorghum depending on the drought scenario. This study analyzed sorghum adaptation mechanisms to early and late vegetative, and the recovery from early vegetative drought stress.

### 4.1. Drought Stress Effect on Plant Phenology, Growth and Development, and Production

In this study, plant phenology was affected through the delay of flag leaf ligulation and flowering. The plants in DS2 conditions, developed most of their leaves (17 out of 19 leaves) before induction of the late vegetative stress. As the presence of all these leaves increased the water requirement of the plant, occurrence of drought stress at this stage was fatal and led to cessation of the plant growth and development, particularly the emergence and development of the very last leaves, which took long to achieve. On the other hand, the plants in DS2 conditions rapidly lost most of their green leaves (an average of 5 leaves for DS2 vs. 13 for WW, remained at flag leaf stage) which could be an adaptation strategy to limit water loss through transpiration. In previous studies, plant phenology, mainly the panicle initiation and flowering time, was also shown to be strongly affected by drought stress [28]. These authors observed with a late vegetative drought stress an increase of the days to flowering and a reduction of the rate of panicle development. This influence on plant phenology was also shown on 21 sorghum genotypes by Rakshit et al. [29], who observed an increase of the days to anthesis under drought stress that was genotype-dependent. In contrast, plant growth and development, above-ground biomass, and grain production were more affected by early drought stress with a decrease of 62% for stem biomass, 55% for leaf biomass, and 62% for panicle dry weight. Previous studies have

also reported a decrement of plant height, above-ground biomass and grain production under early vegetative drought stress conditions [11,30,31]. In this study, the fact that the grain number per panicle was more affected by early vegetative drought compared to late vegetative drought could be explained by the fact that panicle growth starts very early inside the shoot from the growing point differentiation onwards. However, the hundred-seed weight was much more affected by late vegetative drought stress and could be a source limitation for grain filling.

### 4.2. Drought Stress Effect on Plant Physiology

Our results showed that the effect of drought stress was more detrimental on all the physiological parameters during the early vegetative stage compared to the late vegetative stage. The drought stress decreased transpiration and net photosynthetic rates. Meanwhile, the stomatal conductance was also decreased. This could possibly be due to the lowering of $CO_2$ availability for photosynthesis, as a consequence of stomata closure. Many studies reported inhibition of plant photosynthesis due to stomata closure under drought stress to limit water loss through transpiration [32–35]. On the other hand, an increase of intercellular $CO_2$ concentration (Ci) was observed for the droughted treatments, indicating the presence of $CO_2$ in substomatal cavities. However, the $CO_2$ present would not be transferred to the carboxylation sites in the chloroplasts and could explain the inhibition of the photosynthesis activity, which was non-stomatal limitation but appeared to be a mesophyll conductance. Many previous studies reported a non-stomatal limitation of the photosynthesis activity on sorghum, maize, and medicago [36–38]. Mesophyll conductance is reported in many studies to be responsible for carbon fixation reduction in many crops [39–41]. Fv/Fm is used in many studies as index to assess the resistance of the crops to drought stress [37,42,43]. In the present study, the early and late vegetative drought stresses led to the reduction of the maximum yield of PSII (Fv/Fm). Fv/Fm was shown in previous studies to be reduced under drought stress in sorghum [15,44]. The decrement observed in this study could indicate damage caused to the photosystem II reaction center, mainly to the thylakoid membranes. Additionally, a decrease of 14% was also observed for chlorophyll content during early and late vegetative drought stresses. This result is in line with other studies showing negative impact of drought stress on leaf chlorophyll content [15,43,45]. This decrease could indicate chlorophyll degradation due to the damage to the chloroplasts and thus limit photosynthesis [46].

Moreover, plants can develop some protective structures to strengthen their cell wall to cope with drought stress. In previous studies, interlocking marginal lobe (IML) formation was shown to be a drought-tolerance-associated feature [47,48]. In the present study, IML formation was found in both well-watered and drought-stressed sorghum, because sorghum is known for its drought tolerance. In addition, the frequency of the IML increased under drought stress, which again confirmed its role in drought tolerance.

### 4.3. Sorghum Water Use Efficiency and Ability to Recover from Early Drought Stress

Many studies have shown that water use efficiency (WUE) is an important functional index related to plant growth and productivity, and it is used to determine crop optimal water management [49]. In the present study, the WUE differed depending on the plant organ (stem, leaf, grain) from one water treatment to another. WW treatment showed the highest WUE for SFW and SDW, and DS2 better maintained WUE for stem biomass than DS1. For the leaf biomass, DS2 showed the best WUE. As for grain production traits (PFW, PDW, GWP), the highest WUE was obtained with DS1R; however, these differences were not significant. DS2 seemed to better maintain WUE for grain production traits than DS1. These results suggest that the 70% of field capacity applied for the WW treatment was beyond what was needed for optimal WUE; however, the goal of the WW treatment was rather to show the yield potential under non-water-limited conditions. These results are consistent with Bhattarai et al. [50], who studying three irrigations treatments ($I_0$ = 50 mm, $I_1$ = 200 mm, $I_2$ = 350 mm), found that for above-ground biomass at maturity, $I_0$ was

followed by $I_1$, which reached higher WUE than $I_2$. Similar result was also obtained by Abdel-Motagally [51], who studied three water regimes with three grain sorghum genotypes, found that the sorghum plants that received the lower water supply obtained the higher WUE, contrary to those that received the higher water supply. Moreover, Mastrorilli et al. [52], studying a sweet sorghum cultivar named Keller, subjected to early and late vegetative drought stress, found that WUE for the late vegetative stage resulted in higher WUE than well-irrigated and early vegetative drought stress. In short, our results showed the growing point differentiation as a stage at which it is important to avoid drought stress to the plant in order to avoid yield loss and decrease in WUE. Many other studies have reported the growing point differentiation as a critical stage at which drought stress should be avoided and recommended fertilization application prior to this stage and irrigation supply during this stage to increase grain production, especially the number of seeds per head, which is established shortly after this stage [53,54].

Moreover, the recovery test performed in this study showed that the physiological traits such as chlorophyll content, maximum yield of PSII (Fv/Fm), transpiration rate, stomatal conductance to $CO_2$, stomatal conductance to $H_2O$, instantaneous water use efficiency, and photosynthetic rate presented good recovery from early drought stress compared to morphological traits such as plant height and leaf number, which showed low DRI. This result confirms those obtained by Gano et al. [11], who studied early vegetative drought stress on a panel of ten West African sorghum genotypes and showed good recovery index for physiological parameters contrarily to morphological parameters such as plant height and above-ground biomass. They related this photosynthesis recovery ability to the fact that the photosystem reaction center was not irreversibly affected by oxidative damage and also to the plasticity of the genotype to resume with photosynthesis upon rehydration. This reasoning was also given by Devnarain et al. [55] in their study of five African sorghum varieties, which were able to maintain chlorophyll and carotenoid levels upon rehydration after drought stress. Moreover, Martínez-Goñi et al. [12], studying the ability of sorghum to adapt to drought combined with elevated and ambient $CO_2$, observed that after being subjected to drought, sorghum prioritized recovery of its photosynthesis activity upon rehydration mainly by rapidly opening its stomata and increasing the transpiration rate. They also observed that sorghum required more than 7 days of rehydration to fully recover from drought stress.

## 5. Conclusions

Our results show that early vegetative drought was more detrimental on plant vegetative growth, development, and biomass than late vegetative drought stress. Green leaf number and plant leaf area were found to be more affected by DS2 than DS1, and this resulted in delaying the flowering time. The reduction of plant leaf area observed for late vegetative droughted plants is likely an adaptation strategy to limit water loss through transpiration. Nevertheless, plants in DS2 were taller and produced higher stem and leaf biomass compared to DS1. Grain yield was similar for DS1 and DS2, but were outperformed by DS1R and WW. Early drought stress was found to be more deleterious on all the physiological parameters than late vegetative drought stress. The transpiration and photosynthetic rates were shown to decrease at the same time as the stomatal conductance, while an increase of intercellular $CO_2$ concentration limited $CO_2$ uptake and transfer to carboxylation sites to allow photosynthesis. Moreover, the plants recovered quite well from DS1 by increasing the photosynthesis parameters. The WUE for the late vegetative droughted treatment resulted in higher WUE compared to well-irrigated and early vegetative drought treatments. Therefore, it seems important to avoid drought stress at the early vegetative stage (growing point differentiation) where the plants are in rapid growth and development to avoid yield loss and decrease in WUE.

The leaf area reduction to limit transpiration, the reduction of stomatal conductance, and the increase in intercellular $CO_2$ concentration are the adaptative responses observed in

this line. Studying some stress biomarkers and transcriptomic profile will provide further insights into its drought adaptation mechanisms.

**Author Contributions:** T.K.T., S.W., S.V. and R.S. designed the original research questions. T.K.T., Y.B., S.M.A. and S.V. performed the experiments and collected the data. T.K.T. analyzed the data with support from all co-authors. T.K.T. led the writing of the manuscript with contributions from all authors. All authors have read and agreed to the published version of the manuscript.

**Funding:** This research received no external funding.

**Data Availability Statement:** The data are available upon request from the corresponding author (TKT).

**Acknowledgments:** The first author is thankful to DAAD for granting him a post-doctoral fellowship in the framework of the DAAD ClimapAfrica program. The authors are grateful to Mathieu AYENAN for his relevant and constructive comments on the manuscript.

**Conflicts of Interest:** The authors declare no conflict of interest.

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
