# Peer review of "Precision Phenotyping of Agro-Physiological Responses and Water Use of Sorghum under Different Drought Scenarios"

_agronomy, doi:10.3390/agronomy13030722_

Round 1
Reviewer 1 Report
The manuscript is interesting and pleasant to read, it provides useful information on the response of the sorghum crop to different water stress conditions during the growing season.
Allow me to suggest only a few minor revisions.
Enter the field capacity value. What is the value of 70, 35, and 25% of FC? For a better understanding, why not insert the parameter returning the fraction of available water (FC-WP) consumed?
I suggest inserting a Pearson's correlation matrix between investigated crop agro physiological traits under different irrigation conditions. Allows the reader to visually identify any significant correlations.
Line 137, correct "Three to 4" with "Three to four" or "3 to 4"
Line 145, not everyone knows BBCH 30, please write in full
Author Response
Response to Reviewer 1 Comments
Query 1: Enter the field capacity value. What is the value of 70, 35, and 25% of FC? For a better understanding, why not insert the parameter returning the fraction of available water (FC-WP) consumed?
Response:
The field capacity is the water holding capacity of the soil. For the well-watered treatment, we set the watering at 70% of field capacity and this provide optimal water conditions to the plants. It is our control treatment.
By contrast, we applied 25% of field capacity for the water deficit treatment. Also 35% of field capacity was used to maintain the plants in drought stress conditions.
Moreover, it is the fraction of available water consumed, so the transpiration that we plotted in Figure 1 to show the trend of water use for different treatments along the experiment.
Query 2: I suggest inserting a Pearson's correlation matrix between investigated crop agro physiological traits under different irrigation conditions. Allows the reader to visually identify any significant correlations.
Response:
We were also interested in testing the relationship between the agro-physiological parameters assessed. But since the study was done using a single genotype to understand deeply the drought adaption mechanisms, it does not provide any interest in performing the correlation tests. We will not learn any constructive lesson on the relationships since we have very few points.
Query 3: Line 137, correct "Three to 4" with "Three to four" or "3 to 4"
Response:
We have changed it now to “Three to four“
Query 4: Line 145, not everyone knows BBCH 30, please write in full
Response:
Okay. We have written it now fully in the text.
BBCH stands for Biologische Bundesanstalt, Bundessortenamt und CHemische Industrie. It is a scale used to identify the phenological stage of the plant.

Reviewer 2 Report
Dear authors!
Thank you for submitting the manuscript.
Currently, due to global climate change, an important study of the physiology of plant stress, especially salinity, is required. Of particular interest in this regard is the study of C-4 plants, which make up the vegetation in hot arid regions.
In our article, many modern and classical methods for studying the morphometric parameters of plants and determining the photosynthetic system were applied. A lot of work has been done.
I have a number of comments:
1. It is necessary to improve the sound quality in English. The meaning is not always clear. So, for example, line 72, what does the phrase "matic conductivity" mean?
2. Necessary clarification. Line 125. Write the acidity "land from the local field"
3. Line 264-268. You write obvious things that plants in drought conditions develop more slowly than with sufficient watering. This is not a discovery.
4. Fig. 5 is not well done. Pots with plants are close to each other. Poorly visualized. The photo needs to be replaced with a better one.
5. Outdated literature. Only 23% of links in recent years (2020-2022). More recent links need to be added to the list. Moreover, there are a lot of articles on the study of the physiology of drought resistance in plants.
In general, I think that your article is more suitable for a journal in the field of plant physiology research.
Respectfully Yours, reviewer
09 February 2023
Author Response
Response to Reviewer 2 Comments
Dear authors!
Thank you for submitting the manuscript.
Currently, due to global climate change, an important study of the physiology of plant stress, especially salinity, is required. Of particular interest in this regard is the study of C-4 plants, which make up the vegetation in hot arid regions.
In our article, many modern and classical methods for studying the morphometric parameters of plants and determining the photosynthetic system were applied. A lot of work has been done.
I have a number of comments:
Query 1: It is necessary to improve the sound quality in English. The meaning is not always clear. So, for example, line 72, what does the phrase "matic conductivity" mean?
Response.
It is rather “stomatal conductance“ that is written.
Query 2: Necessary clarification. Line 125. Write the acidity "land from the local field"
Response.
The pH (7, alkaline) provided in the text, can give an idea about the acidity of the soil used to fill the pots.
Query 3: Line 264-268. You write obvious things that plants in drought conditions develop more slowly than with sufficient watering. This is not a discovery.
Response.
As we are in Results section, it is the description of the results we obtained regarding the way drought stress affected plant phenology.
Query 4: Fig. 5 is not well done. Pots with plants are close to each other. Poorly visualized. The photo needs to be replaced with a better one.
Response.
Okay. With this picture we would like to show the way early drought stress affected plant morphology, particularly by reducing plant height, appeared leaf number and the size of the leaves. Unfortunately, we don’t have another better picture to replace it.
Query 5: Outdated literature. Only 23% of links in recent years (2020-2022). More recent links need to be added to the list. Moreover, there are a lot of articles on the study of the physiology of drought resistance in plants.
Response.
Most of the papers cited in the present study are really relevant and linked to the study. Even those which were not recent are based on authentic studies focusing on drought stress mainly on sorghum physiology.
In general, I think that your article is more suitable for a journal in the field of plant physiology research.
Respectfully Yours, reviewer

Reviewer 3 Report
Dear Authors,
you have presented a very interesting article covering the range of issues related to the introduction of stress conditions into sorghum cultivation.
In my opinion, the paper is very good and has been written following IMRAD guidelines. You have clearly and comprehensibly described the research methodology and presented constructive results. The results of the experiment were properly presented and illustrated. The discussion was also very substantive.
I would like to ask you to take into account some of the following comments, which will allow a better understanding of the work:
-what was the germination capacity of the seed? did you perform a capacity assessment test beforehand?
-Did you treat the seed before sowing?
-Did you carry out any pesticide treatments?
Although you only present one year of research, I believe that the results of your research are very important from the point of view of improving sorghum varieties. You have indicated specifically what needs to be improved in order to obtain better production results.
Author Response
Response to Reviewer 3 Comments
Dear Authors,
you have presented a very interesting article covering the range of issues related to the introduction of stress conditions into sorghum cultivation.
In my opinion, the paper is very good and has been written following IMRAD guidelines. You have clearly and comprehensibly described the research methodology and presented constructive results. The results of the experiment were properly presented and illustrated. The discussion was also very substantive.
I would like to ask you to take into account some of the following comments, which will allow a better understanding of the work:
Query 1: what was the germination capacity of the seed? did you perform a capacity assessment test beforehand?
Response.
The seeds were from a fresh multiplication trial. Nevertheless, we have proceeded to the germination test and they had all germinated.
Query 2: Did you treat the seed before sowing?
Response.
No. The seeds were not treated before sowing.
Query 3: Did you carry out any pesticide treatments?
Response.
Not specifically for our experiment but phytosanitation treatment for pests regulation is weekly done to the whole greenhouse by the technician.
Although you only present one year of research, I believe that the results of your research are very important from the point of view of improving sorghum varieties. You have indicated specifically what needs to be improved in order to obtain better production results.
